# Evaluation of the Efficacy of Immune and Inflammatory Markers in the Diagnosis of Lacrimal-Gland Benign Lymphoepithelial Lesion

Fuxiao Luan [1,†], Rui Liu [2,†], Jing Li [2], Xin Ge [2], Nan Wang [2], Qihan Guo [2], Yong Tao [1,*] and Jianmin Ma [2,*]

[1] Department of Ophthalmology, Beijing Chaoyang Hospital, Capital Medical University, Beijing 100020, China
[2] Institute of Ophthalmology, Beijing Tongren Eye Center, Beijing Tongren Hospital, Capital Medical University, Beijing 100730, China
* Correspondence: taoyong@bjcyh.com (Y.T.); jmma@ccmu.edu.cn (J.M.)
† These authors contributed equally to this work.

**Abstract:** This study retrospectively analyzes the immune and inflammatory indices of patients with lacrimal-gland benign lymphoepithelial lesion (LGBLEL) in order to screen out reference indices with higher diagnostic efficacy. The medical histories of patients whose diagnoses of LGBLEL and primary lacrimal prolapse were confirmed by pathology between August 2010 and August 2019 were collected. In the LGBLEL group, the erythrocyte sedimentation rate (ESR), C-reactive protein (CRP) level, rheumatoid factor (RF), and immunoglobulins G, G1, G2, and G4 (IgG, IgG1, IgG2, IgG4) were higher ($p < 0.05$) and the expression level of C3 was lower ($p < 0.05$) compared to the lacrimal-gland prolapse group. Multivariate logistic regression analysis showed that IgG4, IgG, and C3 were independent risk factors for predicting LGBLEL occurrence ($p < 0.05$). The area under the receiver operating characteristic (ROC) curve of the prediction model (IgG4+IgG+C3) was 0.926, which was significantly better than that of any single factor. Therefore, serum levels of IgG4, IgG, and C3 were independent risk factors for predicting the occurrence of LGBLEL, and the combined diagnostic efficacy of IgG4+IgG+C3 was the highest.

**Keywords:** lacrimal gland; benign lymphoepithelial lesions; immune marker; inflammatory marker; diagnosis

## 1. Introduction

Benign lymphoepithelial lesion, also known as Mikulicz disease, occurs in the parotid gland, lacrimal gland, and other sites and is characterized by lymphocyte infiltration associated with epithelial hyperplasia [1,2]. Lacrimal-gland benign lymphoepithelial lesion (LGBLEL) is an immune-related inflammatory lesion that is most common in middle-aged women [3,4]. The main manifestations are diffuse enlargement of the bilateral or unilateral lacrimal glands and eyelid swelling. Typical pathological manifestations are diffuse infiltration of lymphocytes and plasma cells in the lacrimal tissue, atrophy and disappearance of glands, and hyperplasia of the fibrous tissue [3,4]. Studies have shown elevated immunoglobulin 4 (IgG4) expression levels in some LGBLEL serum and tissues; therefore, LGBLEL with positive IgG4 expression is considered IgG4-related ophthalmic disease (IgG4-ROD) [5]. The diagnostic criteria for IgG4-ROD depend on the detection of an elevated IgG4+ cell count in a biopsy [6]. However, IgG4 can be increased in xanthogranulomas, Kimura disease, idiopathic orbital inflammation, sarcoidosis, granulomatous polyvasculitis, and mucosa-associated lymphoid tissue lymphomas. Hence, researchers have concluded that elevated IgG4 levels have insufficient sensitivity or specificity for the diagnosis of IgG4-ROD [6–8].

At present, LGBLEL is mainly diagnosed empirically by clinicians through clinical manifestations combined with imaging prior to pathological diagnosis. However, since

LGBLEL lacks characteristic clinical manifestations and is sometimes difficult to distinguish from other lymphoproliferative disease, such as lacrimal-gland lymphoma or an inflammatory pseudotumor of the lacrimal gland, there is a certain risk of misdiagnosis. In view of this, this study analyzes the immune and inflammatory indicators of LGBLEL in order to screen out reference indicators with higher diagnostic efficacy.

## 2. Materials and Methods

For the experimental group, we selected patients who were diagnosed with LGBLEL by histopathology after partial surgical resection of lacrimal-gland tissue at our hospital between August 2010 and August 2019. The control group consisted of patients with primary lacrimal-gland prolapse. The inclusion criteria were ① diagnosis of LGBLEL or primary lacrimal-gland prolapse by histopathology and ② medical history data, including age at diagnosis, sex, duration of disease, treatment and duration of treatment, the interval between diagnosis/treatment and tissue collection, inflammatory markers and immune-related indicators. The exclusion criteria were as follows: ① other systemic rheumatism or immune system disease, ② other lymphoproliferative lesion, such as an inflammatory pseudotumor or lymphoma, ③ secondary lacrimal-gland prolapse caused by another disease, and ④ incomplete medical history data. We specifically included 90 LGBLEL patients and 30 primary lacrimal-prolapse patients based on these inclusion and exclusion criteria. All subjects fully understood the purpose of this study and provided informed consent. The study was supported by the Ethics Committee of Beijing Tongren Hospital Affiliated to Capital Medical University (Beijing, China) in accordance with the principles of the Declaration of Helsinki.

All patients were treated with surgical excision and glucocorticoid therapy. After surgical excision, glucocorticoids (80–120 mg/d) were administered for 3 days and then changed to methylprednisolone tablets (24–28 mg/d). The dosage was reduced by one tablet for 1 to 2 weeks. The course of treatment was 1.5 to 3 months. Besides, peripheral blood samples were taken at the first diagnosis without any treatment for all patients. The tissue samples were taken during the surgery, and all patients were operated within two weeks after first diagnosis.

We recorded patient characteristics such as age at diagnosis, sex, duration of disease, inflammatory markers such as dynamic erythrocyte sedimentation rate (ESR), C-reactive protein (CRP), and angiotensin-converting enzyme (ACE), and immune-related indicators such as antistreptolysin-O (ASO), rheumatoid factor (RF), immunoglobulin M (IgM), complement C3, and IgG and its subtypes (IgG1, IgG2, IgG3, and IgG4). An automatic ESR analyzer was used to detect the level of ESR in patients' peripheral blood. A continuous detection method was used to detect the level of ACE in patients' peripheral blood. The immunonephelometry method was used to detect the other indicators in patients' peripheral blood, including CRP, ASO, RF, complement C3, IgM, IgG, IgG1, IgG2, IgG3, and IgG4.

Immunohistochemical (IHC) staining was utilized to investigate IgG4, IgG, and C3 expression in paraffin-embedded tissues. The diseased tissue sections were dewaxed, incubated at room temperature for 5 to 10 min, washed with distilled water, and soaked in PBS for 5 min. Drops of primary antibody (anti-IgG4, ab271883, Abcam; anti-IgG, ab218427, Abcam; anti-C3, ab200999, Abcam) were added, and the tissue sections were incubated overnight at 4 °C. The tissue was then washed three times with PBS. Biotin-labeled secondary antibody (ab205718, Abcam) was added after 5 min, and the tissue sections were incubated at 37 °C for 30 min. The tissue was washed three more times with PBS, DAB stained, rinsed with water, hematoxylin stained, and mounted [3]. Hematoxylin served as a counterstain, and images were captured with an automatic spectroscopic imaging platform (Vectra II, PerkinElmer, USA). The integrated optical density (IOD) was monitored by Image J software, and the average optical density (AOD = IOD/area) was determined.

We used GraphPad Prism software version 8.0 (GraphPad Software Inc., San Diego, CA, USA) and SPSS version 25.0 (IBM Corp., Armonk, NY, USA) for statis-

tical analysis. Counting data were compared with an $\chi^2$ test or Fisher's exact test. We subjected numerical data to a normality test and the normally distributed data to a *t* test to analyze differences between groups. Independent risk factors were analyzed with logistic regression. $p < 0.05$ was considered indicative of significant differences.

### 3. Results

Ninety LGBLEL patients were enrolled in this study, including 68 females and 22 males. They ranged in age from 18 to 78 years (average = 48.24 ± 11.89 years). Thirty patients had primary lacrimal-gland prolapse, including 25 females and 5 males, and their age range was 18 to 73 years (average = 43.53 ± 12.12 years). There was no significant difference in age distribution between the two groups ($p = 0.07$; Figure 1A). As shown, in the LGBLEL group, the male ratio was 24.44%, and the female ratio was 75.56%. In the control group, the male ratio was 16.67%, and the female ratio was 83.33%. There was no significant difference in sex ratio between the two groups ($p = 0.46$; Figure 1B). In addition, the course of disease was not statistically different between the LGBLEL group (2 to 120 months, average: 21.94 ± 24.47 months) and the control group (3 to 150 months, average: 26.77 ± 32.87 months) ($p = 0.39$; Figure 1C).

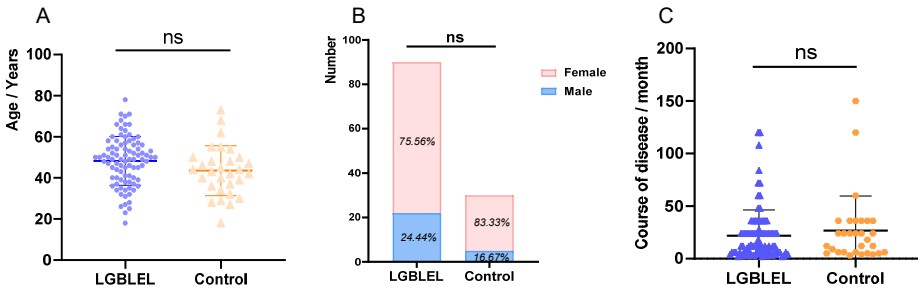

**Figure 1.** Comparative analysis images of general conditions of patients in LGBLEL group and control group. (**A**) Comparative analysis chart of age; (**B**) Comparative analysis chart of gender; (**C**) Comparative analysis chart of course of disease. "ns" refers to no statistically significant difference ($p > 0.05$).

As Table 1 and Supplementary Figure S1 indicate, the average ESR was 17.34 ± 14.10 mm/h in the LGBLEL group and 12.04 ± 6.68 mm/h in the lacrimal-gland prolapse group ($p = 0.0093$). The average CRP level was 2.01 ± 2.89 mg/L in the LGBLEL group and 1.03 ± 1.26 mg/L in the lacrimal-gland prolapse group ($p = 0.0155$). The average RF level of the LGBLEL group was 15.48 ± 28.46 IU/mL, and that of the lacrimal-gland prolapse group was 5.88 ± 3.84 IU/mL ($p = 0.0031$). The average C3 level was 973.77 ± 255.43 mg/L in the LGBLEL group and 1115.49 ± 159.23 mg/L in the lacrimal-gland prolapse group ($p = 0.0012$). The average IgG content in the LGBLEL group was 1531.87 ± 517.47 mg/dl, and that in the lacrimal-gland prolapse group was 1113.79 ± 272.23 mg/dl ($p < 0.0001$). The average IgG1 level was 710.57 ± 287.27 mg/dl in the LGBLEL group and 640.41 ± 156.65 mg/dl in the lacrimal-gland prolapse group ($p = 0.0350$). The average IgG2 level in the LGBLEL group was 589.97 ± 253.84 mg/dl, and that in the lacrimal-gland prolapse group was 449.25 ± 153.76 mg/dl ($p < 0.0001$). Finally, the average IgG4 level was 155.52 ± 197.64 mg/dl in the LGBLEL group and 26.99 ± 20.44 mg/dl in the lacrimal-gland prolapse group ($p < 0.0001$). Each of these sets of results shows a statistically significant difference between the two groups, as indicated by their corresponding *p* values. There were no significant between-group differences in the expression levels of ACE, ASO, IgM, or IgG3 ($p > 0.05$).

We drew a ROC curve for the eight biochemical indicators (ESR, CRP, RF, C3, IgG, IgG1, IgG2, and IgG4) with diagnostic value and calculated the area under the curve (AUC) to evaluate their diagnostic efficiency (Figure 2). RF, CRP, ESR, and IgG1 had low diagnostic efficacy with respective AUCs of 0.684, 0.669, 0.613, and 0.587 ($p < 0.05$, Figure 2A). IgG4, IgG, C3, and IgG2 showed high diagnostic efficacy with AUC values of 0.846, 0.812, 0.729, and 0.710, respectively ($p < 0.05$; Figure 2B,C).

**Table 1.** Biochemical Indicators of the LGBLEL Patients and the Control Group.

| Biochemical Indicators | LGBLEL (n = 90) | Control (n = 30) | t Value | *P* Value |
|---|---|---|---|---|
| ESR (0–20 mm/h) | 17.34 ± 14.10 | 12.04 ± 6.68 | 2.654 | 0.0093 |
| CRP (0–5 mg/L) | 2.01 ± 2.89 | 1.03 ± 1.26 | 2.462 | 0.0155 |
| ACE (33.3 ± 10.2 U/mL) | 33.36 ± 19.92 | 32.59 ± 40.12 | 0.132 | 0.8953 |
| ASO (0–200 IU/mL) | 97.49 ± 118.50 | 88.17 ± 62.35 | 0.5318 | 0.5962 |
| RF (0–20 IU/mL) | 15.48 ± 28.46 | 5.88 ± 3.84 | 3.0400 | 0.0031 |
| C3 (900–1800 mg/L) | 973.77 ± 255.43 | 1115.49 ± 159.23 | 3.374 | 0.0012 |
| IgM (0.4–2.3 g/L) | 1.35 ± 1.89 | 1.26 ± 0.61 | 0.3793 | 0.7052 |
| IgG (751–1560 mg/dL) | 1588.37 ± 530.49 | 1117.86 ± 233.73 | 6.325 | <0.0001 |
| IgG1 (381–930 mg/dl) | 722.04 ± 293.96 | 631.63 ± 156.42 | 2.139 | 0.035 |
| IgG2 (242–700 mg/dl) | 631.17 ± 241.82 | 468.73 ± 161.64 | 4.114 | <0.0001 |
| IgG3 (22-176 mg/dl) | 59.22 ± 39.41 | 50.27 ± 28.68 | 1.339 | 0.1851 |
| IgG4 (4–87 mg/dl) | 174.64 ± 204.74 | 28.82 ± 23.02 | 6.561 | <0.0001 |

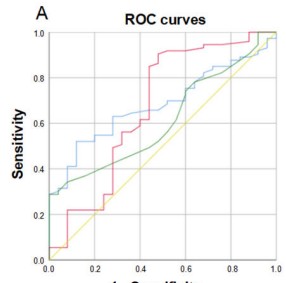 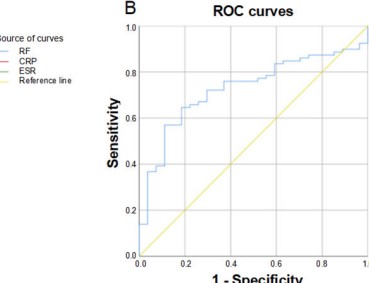 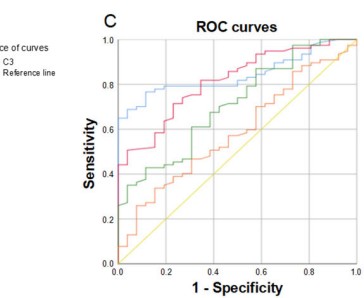

**Figure 2.** ROC curves of different effective indicators. (**A**) ROC curves of RF, CRP, and ESR; (**B**) ROC curve of C3; (**C**) ROC curves of IgG4, IgG, IgG2, and IgG1.

We performed a binary logistic regression analysis on these eight detection indicators, which differed in level between the two groups. The results demonstrate that C3, IgG, and IgG4 were independent predictors of LGBLEL (Table 2, *p* < 0.05), while ESR, CRP, RF, IgG1, and IgG2 were not independent predictors of LGBLEL (Supplementary Table S1, *p* > 0.05). In addition, the prediction model we constructed (IgG4+IgG+C3) had the best prediction accuracy for the diagnosis of LGBLEL. Its ROC AUC reached 0.926, which had the greatest diagnostic value, and was significantly better than that of any single factor (Figure 3). The ROC AUC of IgG4+IgG was 0.892, which signaled high diagnostic efficacy, and exceeded that of any single factor (Figure 3).

**Table 2.** Multifactor Logistic Regression Analysis of Differential Indicators between the LGBLEL patients and the Control Group.

| Indicators | Wald-Value | *p* Value | Correlation | 95% Confidence Interval | |
|---|---|---|---|---|---|
| | | | | Lower Limit | Upper Limit |
| C3 | 4.957 | 0.026 | 0.995 | 0.991 | 0.999 |
| IgG | 7.575 | 0.006 | 1.006 | 1.002 | 1.010 |
| IgG4 | 7.873 | 0.005 | 1.034 | 1.010 | 1.058 |
| Constant | 1.369 | 0.242 | 0.061 | - | - |

Since the AUC of IgG4+IgG+C3 in peripheral blood was the largest, IgG4, IgG, and C3 are usually more easily enriched in diseased tissues than in peripheral blood, we further performed IHC staining with anti-IgG4, anti-IgG and anti-C3 antibodies in the lacrimal-gland tissues of patients with LGBLEL and primary lacrimal prolapse to validate the different expressions of these biomarkers. As shown in Figure 4A–F, the IHC staining of IgG4, IgG, and C3 in the LGBLEL patients' lacrimal-gland tissues revealed extensive positive expressions, whereas it displayed negative expressions in patients with primary lacrimal-gland

prolapse. The quantitative findings indicated statistically significant differences between the two groups by ImageJ (*n* = 5, *p* < 0.0001; Figure 4G–I).

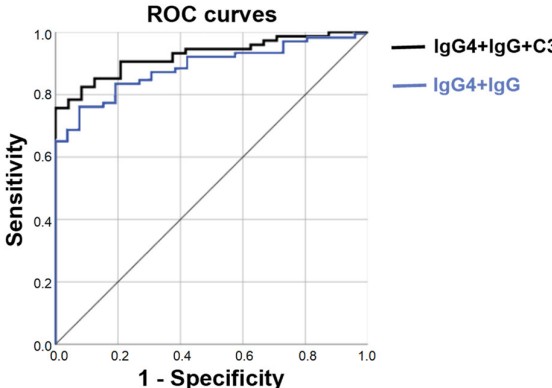

**Figure 3.** ROC curves of different prediction models. The black line represents the ROC curve of IgG4+IgG+C3 (AUC = 0.926), and the blue line represents the ROC curve of IgG4+IgG (AUC = 0.892).

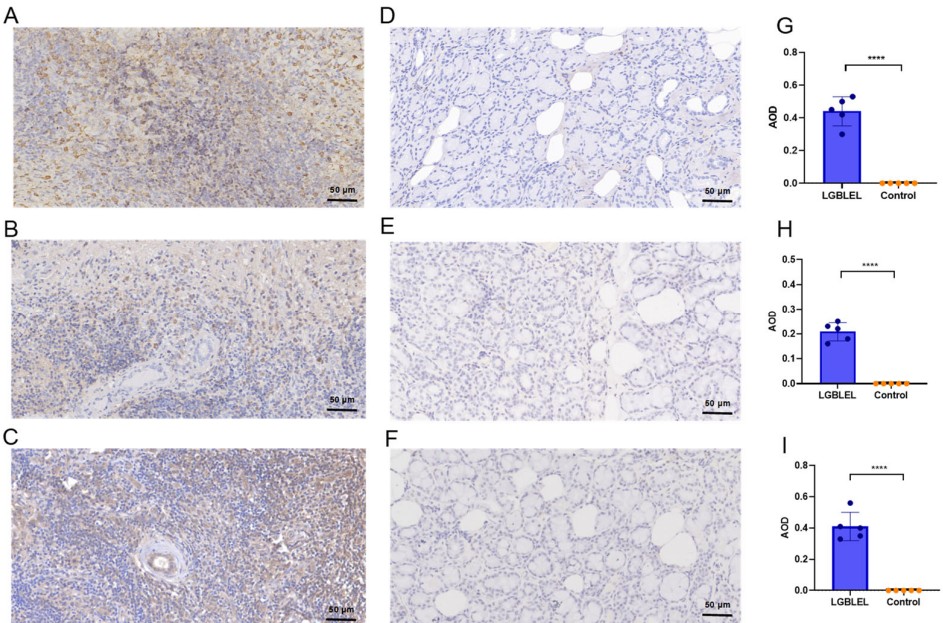

**Figure 4.** Expression of IgG4, IgG, and C3 in the tissues of patients with LGBLEL and primary lacrimal prolapse. (**A**–**C**) The IHC staining representative images of IgG4, IgG, and C3 positive expressions in the lacrimal-gland tissues of LGBLEL patients, respectively; (**D**–**F**) The IHC staining representative images of IgG4, IgG, and C3 negative expressions in the lacrimal-gland tissues of patients with primary lacrimal prolapse, respectively; (**G**–**I**) Quantitative analyses of IgG4, IgG, and C3 expressions by IHC, respectively (*n* = 5, *p* < 0.0001). Note: "****" stands for a statistically significant difference.

## 4. Discussion

Lacrimal-gland benign lymphoepithelial lesion is an immune-related inflammatory lesion whose pathogenesis might be related to the FcepsilonRI, receptor (FcεRI), B cells, T cells, and complement system signaling pathways [3,9,10]. With economic and societal developments, LGBLEL has become a more common orbital inflammatory disease. It is characterized by a long disease course and a tendency of recurrence. Currently, there are no unified diagnostic criteria or treatment plan. Most LGBLEL patients present with lacrimal-gland enlargement, eyelid swelling, or eyelid redness, among other symptoms. Clinicians rely only on clinical manifestations and imaging examinations, and LGBLEL

is easily mistaken for other diseases. Therefore, this study was conducted from the perspectives of immunity and inflammation to find indicators that can assist in diagnosis and differentiation.

Lacrimal-gland benign lymphoepithelial lesion is considered IgG4-ROD. Research suggests that serum IgE, RF, C3, C4, IgG1, IgG2, and IgG4 are factors that affect the occurrence and recurrence of IgG4-ROD [11–13]. At present, these indicators have not been applied to the diagnostic reference criteria of IgG4-ROD. According to these criteria, the expression level of IgG4 is still the main reference index for diagnosis of this disease [14,15]. However, since there is evidence that IgG4 is positively expressed in a variety of diseases, elevated serum IgG4 is not considered specific to LGBLEL [16]. To improve the sensitivity of diagnosis and differentiation, Detiger et al. have proposed the use of additional IHC staining of IgG2[+] plasma cells and the histological IgG2/IgG4 ratio in diagnosis of IgG4-ROD [17]. Furthermore, Chan et al. have reported that the sensitivity, specificity, and accuracy of serum IgG2 critical value > 5.3 g/L were 80%, 91.7%, and 0.90, respectively, for diagnosis of orbital IgG4-ROD [18]. Arora et al. have suggested that an increased histological IgG4/IgG ratio is helpful to diagnose IgG4-ROD [19]. In this study, we sought to support disease diagnosis and differentiation by screening for simpler serological indicators.

Based on the results of this study, we propose that the clinical diagnosis of LGBLEL will have high accuracy depending on the clinical manifestations and imaging characteristics combined with an increase in serum IgG4 and IgG and a decrease in C3. At present, the specific mechanism of IgG and IgG4 in LGBLEL pathogenesis is not clear, although it might be related to the activation of the complement system. Studies have shown that complement activation occurs via three pathways: classical, mannose-binding lectin and alternative pathways [20]. After complement activation, proinflammatory mediators are released, triggering an inflammatory response and immunoreaction [21]. IgG1 and IgG4 can activate the complement by activating C1q; IgG1~IgG3 through activation of C1q, cause C1q configuration changes and exposure of CH2/CH3 ribbon binding sites, activate the complement in the classical way, and induce the release of C3a, C5a, and C5b-9. However, IgG4 can also activate the complement through the bypass pathway [22,23]. C3, C5, and C9 all play important roles in both the classical and bypass pathways of the complement. Previously, through transcriptome sequencing, our team preliminarily revealed the involvement of the complement system in the pathogenesis of LGBLEL [10]. The above conclusions support the results of this study. IgG, IgG4, and C3 could be independent predictors of LGBLEL and involved in the disease's pathogenesis via activation of the complement system.

This study has some limitations. The preoperative empirical glucocorticoid administration might have affected the expression of serological indicators to some extent, which could have led to deviations in the results. Moreover, lacrimal-gland prolapse can have a variety of causes and is often accompanied by infiltration by a small number of inflammatory cells. To reduce these influencing factors, all patients with lacrimal-gland prolapse who were recruited for this study had normally sized lacrimal glands, as observed in magnetic resonance imaging scans. In addition, during case selection, we excluded any cases with pathological manifestations of inflammatory-cell infiltration to avoid the influence of this factor.

## 5. Conclusions

In summary, based on a clinical review, this study combined clinical manifestations of LGBLEL with laboratory detection indicators and used patients with primary lacrimal-gland prolapse as the control group to detect laboratory indicators, such as ESR, CRP, RF, C3, and IgG and its subtypes, in peripheral blood. We then screened out indicators with diagnostic value and evaluated the diagnostic efficacy of each. Our results identify IgG4+IgG+C3 as the prediction model with the highest diagnostic efficacy.

**Supplementary Materials:** The following supporting information can be downloaded at: https://www.mdpi.com/article/10.3390/cimb45030129/s1, Figure S1: Comparative analysis images of immune and inflammatory indicators between LGBLEL and control groups; Table S1: Multifactor Logistic Regression Analysis of Other Differential Indicators between the LGBLEL patients and the Control Group.

**Author Contributions:** F.L. and R.L. analyzed and wrote the manuscript; J.L., X.G. and N.W. helped collect the data; Q.G. contributed to data analysis; Y.T. and J.M. read and critiqued the manuscript. All authors have read and agreed to the published version of the manuscript.

**Funding:** Supported by National Natural Science Foundation of China (82070948), Beijing Hospitals Authority' Ascent Plan (DFL20190201 and No. DFL20220301), Natural Science Foundation of Beijing (7222025), and Shunyi District Beijing Science and Technology Achievements Transformation Coordination and Service Platform Construction Fund (SYGX202010).

**Institutional Review Board Statement:** The study was conducted in accordance with the Declaration of Helsinki, and approved by the Institutional Review Board (or Ethics Committee) of NAME OF INSTITUTE (TRECKY2019-093 and 2019-08-01).

**Informed Consent Statement:** This article does not include the patients' names, portraits, or other private information. Informed consent was obtained from each patient for the publication of this article and any accompanying images.

**Data Availability Statement:** Not applicable.

**Conflicts of Interest:** All authors declare no conflict of interest.

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
