# Peer review of "Evaluation of the Efficacy of Immune and Inflammatory Markers in the Diagnosis of Lacrimal-Gland Benign Lymphoepithelial Lesion"

_cimb, doi:10.3390/cimb45030129_

Round 1
Reviewer 1 Report
cimb-2172796
In this work, the author explored the immune and inflammatory indicators for LGBLEL screening. The topic is of great significance and the data have practical clinical implications. The manuscript is well-written and can be accepted after minor revision. From the perspective of academic criticism, several concerns need to be addressed to further improve the quality of this manuscript, as appended below.
-
In addition to Figure 1, a table listing all the patient information (age, gender, diagnosis) should be added to the supplementary materials.
-
The “ns” label in Figure 1B should be within each group (one “ns” for LGBLEL and one for control). If the author wants to emphasize the male/female ratio in two groups, then please just show the ratio with the statistical analysis.
-
Images/figures with higher resolution should be used in Figures 3 and 5. Also, an IHC image with higher magnification would be better for the presentation in Figure 5.
-
It would be better for the presentation if the two ROC curves were combined in the same figure in Figure 4.
Author Response
Dear Reviewer,
Thank you for your comments concerning our manuscript entitled “Evaluation of the efficacy of immune and inflammatory markers in the diagnosis of lacrimal-gland benign lymphoepithelial lesion” (Manuscript No. cimb-2172796). Your comments are all valuable and very helpful for revising and improving our paper. We respect your earnest work and have carefully studied the comments before making corrections that we hope meet with your approval. We have not listed all the changes made here, but these are all marked in the revised manuscript. The main corrections and the responses to your comments are as follows:
Comments and Suggestions for Authors cimb-2172796
1. In addition to Figure 1, a table listing all the patient information (age, gender, diagnosis) should be added to the supplementary materials.
Our response: Thank you for your comments. We added a table including all the patient information (age, gender, diagnosis) to the supplementary material ( an excel file named ‘patient information’).
2. The “ns” label in Figure 1B should be within each group (one “ns” for LGBLEL and one for control). If the author wants to emphasize the male/female ratio in two groups, then please just show the ratio with the statistical analysis.
Our response: Thank you for your comments. According to your suggestions, we make the following modifications to the question of the male/female ratio in two groups:
Ninety LGBLEL patients were enrolled in this study, including 68 females and 22 males. They ranged in age from 18 to 78 years (average, 48.24±11.89 years). Thirty patients had primary lacrimal-gland prolapse, including 25 females and 5 males; their age range was 18–73 years (average, 43.53±12.12 years). There was no significant difference in age distribution between the two groups (P =0.07; Fig. 1A). As is shown, the male ratio is 24.44% and the female ratio is 75.56% in LGBLEL group. And the male ratio is 16.67% and the female ratio is 83.33% in control group. There was also no significant difference in sex ratio between the two groups (P = 0.46; Fig. 1B). In addition, the course of disease was not statistically different between the LGBLEL group ( 2 to 120 months, average: 21.94±24.47 months) and the control group ( 3 to 150 months, average: 26.77±32.87 months) (P = 0.39; Fig. 1C).
Figure 1. Comparative analysis images of general conditions of patients in LGBLEL group and control group. (A) Comparative analysis chart of age; (B) Comparative analysis chart of gender; (C) Comparative analysis chart of course of disease. “ns” refers to no statistically significant difference (P > 0.05).
3. Images/figures with higher resolution should be used in Figures 3 and 5. Also, an IHC image with higher magnification would be better for the presentation in Figure 5.
Our response: Thank you for your comments. In line with your recommendations, we have modified and adjusted the pictures in Figure 3 and Figure 5. In addition, we had IHC images of IgG, IgG4 and C3 with higher magnification in Figure 5.
Figure 3. ROC curves of different effective indicators. (A) ROC curve of RF、CRP and ESR; (B) ROC curve of C3; (C) ROC curve of IgG4、IgG、IgG2 and IgG1.
Figure 5. Expression of IgG4, IgG, and C3 in the tissues of patients with LGBLEL and primary lacrimal prolapse. (A-C) The IHC staining representative images of IgG4, IgG, and C3 positive expressions in the lacrimal-gland tissues of LGBLEL patients, respectively; (D-F) The IHC staining representative images of IgG4, IgG, and C3 negative expressions in the lacrimal-gland tissues of patients with primary lacrimal prolapse, respectively; (G-I) Quantitative analyses of IgG4, IgG, and C3 expressions by IHC, respectively (n = 5, P < 0.0001).
4. It would be better for the presentation if the two ROC curves were combined in the same figure in Figure 4.
Our response: Thank you for your comments. According to your suggestion, we have revised the picture in Figure 4, as follows:
Figure 4. ROC curves of different prediction models. The black line represented the ROC curve of IgG4+IgG+C3 (AUC = 0.926), and the blue line represented the ROC curve of IgG4+IgG (AUC = 0.892).

Reviewer 2 Report
This study, aimed at identifying specific diagnostic biochemical indicators of lacrimal gland benign lymphoepithelial lesions (LGBLEL), retrospectively compared serum levels of a set of immunological proteins between cases and unrelated-disease controls. It found levels of three of the proteins, combined, to be the most efficient indicator of the disease. Levels of one of these proteins were also determined by immunohistochemistry.
It is a novel study and addresses a clinically relevant and important question. As the diagnostic criteria used for diagnosis of LGBLEL are not very specific, more specific criteria with better predictive value would be useful. Several issues have been noted with the study and its presentation, which must be addressed before considering for publication.
The Introduction must adequately acknowledge the literature on this disease and statements must be supported by relevant references.
The inclusion and exclusion criteria for recruitment of cases and of controls must be separately indicated. What is meant by “complete medical history” needs description. Specify which ELISA were used for analysing each protein. Was immunohistochemistry performed on specimens collected for pathology or research? Either refer to a published method or briefly describe the method of immunohistochemistry.
Besides age and sex, present other characteristics of the study cohort, including age at diagnosis, duration of disease, treatment and duration of treatment, and interval between diagnosis/treatment and tissue collection for analyses. It is also important to determine whether any of the characteristics have a confounding effect of the results.
Can the authors also perform immunohistochemistry of IgG and C3 and present that data?
For completeness of information, can they also present the outcome of logistic regression analysis of the proteins that were not found to be independent predictors of the disease, in a supplementary table?
Figure 2 presents the same data as presented in Table 2 and therefore seems redundant. Indicate the normal range of each protein in Table 2 rather than in the text. Order of Tables 1 and 2 must be changed to be consistent with the text and define abbreviations used in a table. Figure legends should be improved to be more informative without being repetitive.
It would be more appropriate to refer to the analysed proteins as “Biochemical indicators” rather than “Laboratory indicators”.
In the Discussion, instead of repeating the results (3rd paragraph), include an in-depth discussion about the mechanism of activation of the complement system in light of the present and reported studies.
A few grammatical errors were noted. They must be corrected.
Author Response
Dear Reviewer,
Thank you for your comments concerning our manuscript entitled “Evaluation of the efficacy of immune and inflammatory markers in the diagnosis of lacrimal-gland benign lymphoepithelial lesion” (Manuscript No. cimb-2172796). Your comments are all valuable and very helpful for revising and improving our paper. We respect your earnest work and have carefully studied the comments before making corrections that we hope meet with your approval.
Please see the attachment.
